# SPACE: Single-round Participant Amalgamation for Contribution Evaluation in Federated Learning

**Yi-Chung Chen**
National Taiwan University
r10942081@ntu.edu.tw

**Hsi-Wen Chen**
National Taiwan University
hwchen@arbor.ee.ntu.edu.tw

**Shun-Gui Wang**
National Taiwan University
r11921099@ntu.edu.tw

**Ming-Syan Chen**
National Taiwan University
mschen@ntu.edu.tw

## Abstract

The evaluation of participant contribution in federated learning (FL) has recently gained significant attention due to its applicability in various domains, such as incentive mechanisms, robustness enhancement, and client selection. Previous approaches have predominantly relied on the widely adopted Shapley value for participant evaluation. However, the computation of the Shapley value is expensive, despite using techniques like gradient-based model reconstruction and truncating unnecessary evaluations. Therefore, we present an efficient approach called Single-round Participants Amalgamation for Contribution Evaluation (SPACE). SPACE incorporates two novel components, namely *Federated Knowledge Amalgamation* and *Prototype-based Model Evaluation* to reduce the evaluation effort by eliminating the dependence on the size of the validation set and enabling participant evaluation within a single communication round. Experimental results demonstrate that SPACE outperforms state-of-the-art methods in terms of both running time and Pearson's Correlation Coefficient (PCC). Furthermore, extensive experiments conducted on applications, client reweighting, and client selection highlight the effectiveness of SPACE. The code is available at https://github.com/culiver/SPACE.

## 1 Introduction

With the rapid growth of mobile services, the combination of rich user interactions and robust sensors means they are able to access an unprecedented amount of data. While conventional machine learning approaches require centralizing the training data in a single machine or a data center, which may limit the scalability of machine learning applications, Federated Learning (FL) [36] has been proposed to leave the training data distributed on the client side and then aggregate locally computed updates, e.g., the gradient of the neural network, to a central coordinating server, without accessing the raw data. A line of subsequent efforts has significantly improved the efficiency [1, 15, 30, 62, 70, 73], and robustness [17, 38, 63] of FL. Moreover, such distributed training paradigm has been adapted to client personalization [8, 27, 40, 41], and heterogeneous model architecture [10, 75], making it even more practical.

Recently, there has been increasing interest in evaluating the participant's contribution to FL for a fair incentive mechanism [12, 29, 33, 45, 46, 55, 65, 69, 71, 72] based on the participant contribution in attracting data owners. Additionally, deciding the participant weights [54, 57] during the training process based on their contributions can accelerate the convergence of the training process but also enhance the model's robustness by alleviating the negative effect from the confused or even incorrect

37th Conference on Neural Information Processing Systems (NeurIPS 2023).

data. Moreover, if the communication budget is limited, participant contributions can be utilized as a reference to select important participants [7, 11, 13, 53, 56, 73], reducing communication costs while maintaining the model's performance.

One straightforward method to evaluate participant contribution is to assess their individual performance independently. For instance, the accuracy of the local model has been employed as a measure [39], as well as the consistency between the local and global models, indicating their similarity to the consensus of the coalition [45, 72]. However, these approaches do not fully consider the impact of client cooperation. To consider cooperation, some studies have adopted the influence function [21] from centralized data evaluation to assess participant contribution [57, 64]. However, calculating the influence function involves clients sending their locally computed Hessian matrices to the server, leading to unacceptable client computation and communication costs. On the other hand, researchers have explored using the Shapley value [42] in cooperative game theory to measure participant contributions. While the exact computation of the Shapley value requires enumerating all possible client combinations, Song et al. [48] approximate the target model with gradient updates to avoid model retraining. To further speed up the computation, Wei et al. [59] and Liu et al. [32] propose truncation techniques, eliminating unnecessary model reconstruction and evaluation.

Despite the aforementioned efforts, the approximation cost of the Shapley value can still be prohibitively high for the following two reasons. **i) Multi-Round Training:** FL algorithms, such as FedAvg, involve multiple communication rounds wherein models are transmitted between the server and clients, and local training takes place on the client side before achieving model convergence. Previous studies [32, 48, 57–59] compute participant contributions for each communication round and aggregate them to obtain the overall participant contribution. Consequently, the communication and computation efforts escalate with an increase in the number of communication rounds. **ii) Dataset-Dependent Evaluation:** As demonstrated in [32], the evaluation of models constitutes the primary computational time component required for calculating the Shapley value. Even with the adoption of permutation sampling approximation, the complexity of evaluating per-round contributions depends on the size of the validation dataset on the server. As the validation set continually grows, the evaluation process becomes computationally unacceptable.

In this paper, we propose *Single-round Participants Amalgamation for Contribution Evaluation (SPACE)* to efficiently calculate the participant contribution in FL. Specifically, we analyze the data distribution between the server and the client dataset. If the server and client distributions are similar, the client dataset proves helpful in correctly classifying the data in the validation set of the server. While previous works [37, 51, 52, 68] have shown that prototypes, i.e., the embedding vectors representing each class, can be employed as a privacy-preserving approach to compare data distribution, we propose *Prototype-based Model Evaluation*, which evaluates the model performance by measuring the similarities between the local and server prototypes. Therefore, the distribution of the validation set can be constructed and thus eliminates the dependency of evaluation complexity on the size of the validation set.

To derive meaningful prototype embeddings, we propose a novel approach called *Federated Knowledge Amalgamation*, wherein client models can be viewed as multiple distributed teacher models, and their knowledge can be distilled to educate the server model simultaneously. Thus, federated knowledge amalgamation enables us to distill knowledge from all local models in just one communication round to reduce the time for prototype construction. The educated model is distributed to each participant, who then constructs prototypes based on their local data. We then evaluate the model performance by measuring the similarities between server and client prototypes. Besides, unlike previous works that adopt the model performance as the utility function, we modify the utility function by incorporating a logistic function. This adjustment aims to reflect users' satisfaction better, thereby enhancing the rationality of the utility function in real-world scenarios. Extensive results demonstrate the advantage of SPACE in multiple applications, including contribution evaluation, client re-weighting, and client selection.

The contributions of this paper are summarized as follows:

- We propose SPACE, an efficient and effective approach for client contribution evaluation in an FL setting, which significantly reduces the required communication and computational cost and exhibits outstanding performance in terms of Pearson's correlation coefficient (PCC).

- We introduce *Prototype-based Model Evaluation*, which effectively removes the computation dependency on the size of the validation set. To efficiently establish a robust embedding space for

building prototypes, we introduce *Federated Knowledge Amalgamation* that enables the aggregation of embedding space information from local models in an FL setting, all within a single round of communication.

- Extensive experiments have been conducted to show that SPACE outperforms previous state-of-the-art methods in terms of both running time and PCC. SPACE consistently exhibits exceptional performance in applications including client reweighting and selection.

## 2 Related Works

**Contribution evaluation in FL.** Client contribution evaluation in federated learning (FL) has gained significant attention for its potential applications, including attack detection [14, 31], debugging [25, 26], incentive mechanisms [33, 45, 46, 48, 55, 65, 72], and convergence acceleration [54, 57]. This concept draws inspiration from centralized data evaluation tasks [16, 19]. Recent works have proposed methods tailored for FL. Pandey et al. [39] consider the relative accuracy of the local model as individual contribution. Zhang et al. [72] calculate the cosine similarity between local gradients and a unit vector pointing towards the optimal model as contributions. Shi et al. [45] measure contribution based on the cosine similarity between local and global gradients. However, these methods do not fully capture the impact of cooperation. To address this limitation, the Shapley value is widely adopted for evaluating client contribution in FL due to its desirable properties, including efficiency, symmetry, linearity, and null player property. However, directly computing the Shapley value incurs exponential computational costs involving model retraining and inference. To alleviate this burden, Song et al. [48] reconstruct the target model using gradient updates to avoid model retraining. Wang et al. [58] enhance the evaluation using permutation sampling and a group testing approach. Additionally, Wei et al. [59] and Liu et al. [32] incorporate truncation techniques to reduce computation costs. Despite these efforts, approximating the Shapley value still requires significant computation.

**Knowledge Amalgamation.** Knowledge amalgamation (KA), a variant of knowledge distillation [18], involves a setup with one student model and multiple teacher models. KA has been widely adopted in prior works [5, 20, 34, 43, 44, 61, 66, 67] due to its desirable properties. Shen et al. [43] introduced KA, where compact feature representations are learned from teachers and used to update the student model. Shen et al. [44] proposed a strategy that extracts task-specific knowledge from diverse teachers to create component networks, which are then combined to construct the student network. Luo et al. [34] introduced a common feature learning scheme in heterogeneous network settings, transforming the features of all teachers into a shared space. Xie et al. [61] proposed a related approach applying KA to transfer prior knowledge from multiple pre-trained models. However, their focus is on knowledge transfer from given pre-trained models, while our approach is the first to employ KA within a conventional FL framework. In our setting, client models can be viewed as distributed teacher models, and their knowledge can be distilled to educate the server model.

## 3 Preliminary

Since federated learning aims to construct a unified, global machine-learning model to collect data information located on numerous remote devices (client), the objective is to develop this model while ensuring that the data generated by each device is processed and stored locally, with only intermediate updates being sent to a central server at regular intervals. The primary objective is usually to minimize the objective function as follows. Let $\mathcal{N}$ denote a set consisting of $n$ clients,

$$\theta^* = \arg\min_{\theta} \sum_{i}^{n} p_i L_i(\theta),  \tag{1}$$

where $p_i \geq 0$ and $\sum_i p_i = 1$. $L_i$ is the objective function for the $i$-th client. Generally, each client may utilize the same objective function, such as the empirical risk, which can be defined as follows.

$$L_i = \sum_{(x_{i,j}, y_{i,j}) \in \mathcal{D}_i} l(\theta, x_{i,j}, y_{i,j}),  \tag{2}$$

where $\mathcal{D}_i$ is the dataset in $i$-th client. $p_i$ specifies the relative impact of each device, such as the uniform weighting $\frac{1}{n}$ or depending on the number of samples in each dataset $\frac{|\mathcal{D}_i|}{\sum_i^n |\mathcal{D}_i|}$.

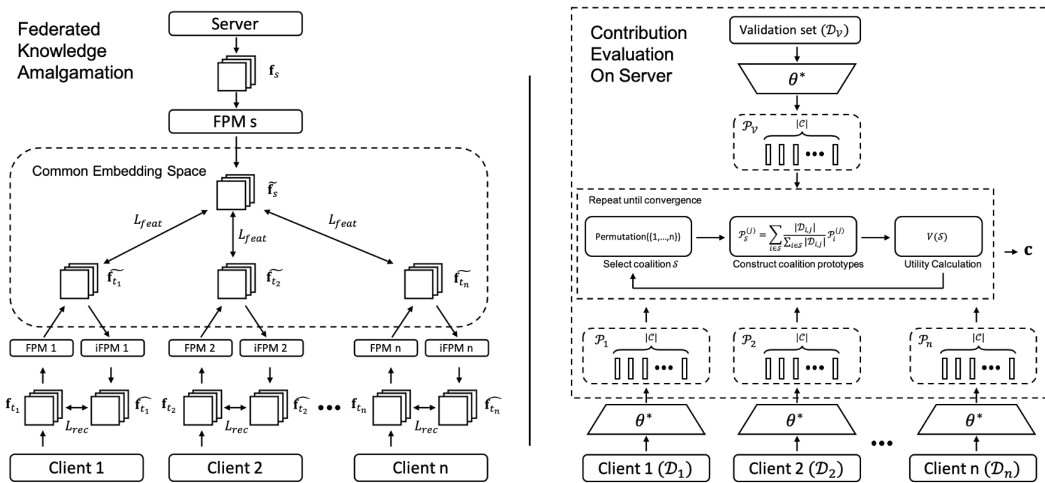

Figure 1: Overall architecture of SPACE. Initially, the clients train their respective local models using their local datasets. Subsequently, the locally optimized models are frozen and serve as teacher models to educate the student model on the server by Federated Knowledge Amalgamation. Then, the educated model $\theta^*$ is distributed to all clients, who then utilize it to construct their corresponding prototypes $\mathcal{P}_i$. These prototypes are then uploaded to the server. The server employs the Prototype-based Model Evaluation approach to assess the model's performance and determine the contribution $\mathbf{c}$ of each participant.

Despite early efforts to enhance the efficacy of federated learning, many data owners hesitate when it comes to partaking in data federation owing to apprehensions surrounding the possibility of inequitable remuneration. Furthermore, in federated learning, privacy concerns preclude the server from directly accessing unprocessed data stored on client devices. Consequently, there is a pressing need for a privacy-preserving federated contribution evaluation method that ensures equitable distribution of rewards.

Specifically, the evaluation of participant contributions can be abstracted into cooperative game scenarios. Given a cooperative game $(\mathcal{N}, V)$, where $\mathcal{N}$ denotes the set of $n$ participants and $V(\cdot)$ is a utility function defined as $V : 2^n \to \mathbb{R}$, which assigns a value to a coalition $\mathcal{S} \in \mathcal{N}$, the objective of payoff allocation is to determine a payoff allocation vector, denoted by $\mathbf{c} = (c_1, ..., c_n)$, where $c_i$ represents the payoff for the $i$-th participant in $\mathcal{N}$.

The idea of cooperative games has been extensively researched [2, 3], with the Shapley value being the most commonly used metric due to its existence and uniqueness in a cooperative game. Additionally, the Shapley value is characterized by a set of desirable axiomatic properties, including (i) efficiency, (ii) symmetry, (iii) null player, and (iv) linearity, which make it a desirable approach for evaluating contributions. The Shapley value is defined as follows,

$$\phi_i(\mathcal{N}, V) = \sum_{S \in \mathcal{N} \setminus \{i\}} \frac{|S|!(n - |S| - 1)!}{n!}(V(S \cup \{i\}) - V(S)). \tag{3}$$

$\mathcal{S}$ denotes the subset of participants from $\mathcal{N}$. The Shapley value can be seen as the average marginal gain that a participant joins the coalitions. To accelerate the approximation, the Monte Carlo estimation method for approximating Shapley values from [16] is adopted,

$$\phi_i = \mathop{\mathbb{E}}_{\pi \sim \Pi}[V(S_\pi^i \cup \{i\}) - V(S_\pi^i)], \tag{4}$$

where $S_\pi^i$ is the set of data from FL participants joining before $i$ in the $\pi$-th permutation of the sequence of FL participants. The calculated Shapley values are clipped to be non-negative to follow the non-negative assumption as in [3].

## 4 Method

Shapley value is a widely used approach for evaluating participant contributions in federated learning due to its desirable properties. However, despite the proposal of various acceleration techniques

such as Monte Carlo sampling, gradient-based model reconstruction, and truncation, the computation of the Shapley value remains prohibitively slow due to the following two key challenges **Multi-Round Training**: multiple communication rounds are required for model convergence, and **Dataset-dependent Evaluation**: the evaluation of model performance depends on the size of the validation set on the server. Thus, we propose the Single-round Participant Amalgamation for Contribution Evaluation (SPACE) approach, which analyzes participant contributions using only one communication round. The key idea behind SPACE is to compare the distribution between the local dataset and the validation dataset directly via prototype embeddings of the local dataset and the validation dataset [37, 51, 52, 68]. To establish robust prototype embeddings, we introduce Federated Knowledge Amalgamation to distill feature information from all locally optimized models to the server model simultaneously, which enables the server model to learn information in a single communication round. Once the prototypes are obtained, we evaluate the performance of coalition $\mathcal{S}$ by comparing the similarities between the prototypes formed by clients in coalition $\mathcal{P}_\mathcal{S}$ and prototypes formed by validation set $\mathcal{P}_\mathcal{V}$. Using prototypes eliminates the need to iterate over all samples in the validation set, resulting in faster performance evaluation. Figure 1 shows the architecture of SPACE.

### 4.1 Federated Knowledge Amalgamation

As previously mentioned, the SPACE framework utilizes prototypes to assess participant contributions. Therefore, employing a semantically meaningful deep model is essential to ensure the extraction of expressive features from data samples. One straightforward approach is to utilize the converged model achieved through federated training, which encompasses training with all the local data. However, the convergence of federated learning algorithms like FedAvg necessitates numerous rounds of local-global communication and local training. Consequently, the evaluation of contributions can only be conducted after extensive training, resulting in excessively high costs and impeding the utilization of participant contributions in applications like client selection.

In contrast to conventional knowledge distillation tasks, which typically involve a single student model and a single teacher model, knowledge amalgamation techniques are employed to entail one student model and multiple teacher models. Furthermore, in federated learning, the client models can be considered teacher models that acquire knowledge from their local datasets. Consequently, the application of knowledge amalgamation facilitates the construction of an expressive embedding space for federated learning.

SPACE first distributes the global model $\theta$ to each client and each client implements empirical risk minimization to obtain a locally optimized model $\theta_i$. The knowledge amalgamation is then applied to transfer the knowledge to the initial global model $\theta$ with the locally optimized models serving as teacher models and the server validation dataset for distillation. To simultaneously distill knowledge from all the teacher models, the features of both the student and teacher models must be projected to common feature space for the feature alignment.

Therefore, we use feature projector modules (FPM) to project the intermediate feature of models to the common feature space. Since the data distributions of clients are different, the projections from the original feature space to the common feature space are also different. Therefore, each model, including the student model and teacher models, would be assigned an individual learnable FPM for projection. After projection, the student model then starts to learn information from teacher models by minimizing the feature distillation loss $L_{feat}$, which is the L1 distance between the projected student features $\tilde{\mathbf{f}}_s$ and the projected teacher features $\tilde{\mathbf{f}}_{t_i}$. However, due to the differences in local data distribution, the teacher models are not equally important, and the loss should be weighted differently depending on the sample used for knowledge amalgamation. We define the weighting of each teacher model $w_i$ by its confidence in predicting the class of data sample. Specifically, given a data sample $d$ with its class as $j$ for distillation, the weighting of each teacher model is the softmax of the $j$-th logit of its prediction $\mathcal{F}(\theta_i, d)$, which can be written as follows:

$$L_{feat} = \sum_i w_i |\tilde{\mathbf{f}}_s - \tilde{\mathbf{f}}_{t_i}|, \ w_i = \frac{e^{\mathcal{F}(\theta_i, d)_j}}{\sum_i e^{\mathcal{F}(\theta_i, d)_j}} \tag{5}$$

To prevent the FPM from projecting all the features to a constant vector, which is a trivial optimal solution for the feature distillation loss, reconstruction loss is proposed to ensure the projected feature contains the knowledge in the original feature space. For each teacher model, we apply an inverse feature projector module (iFPM) to reconstruct features in the original feature space, which

projects $\tilde{\mathbf{f}}_{t_i}$ to the reconstructed feature $\hat{\mathbf{f}}_{t_i}$. The reconstruction loss is defined as follows:

$$L_{rec} = \sum_i |\mathbf{f}_{t_i} - \hat{\mathbf{f}}_{t_i}| \tag{6}$$

where $\mathbf{f}_{t_i}$ is the original feature of $i$-th client before projection. Aside from feature distillation, we also use prediction distillation to enhance the overall amalgamation performance. For prediction distillation, the student model only learns from the teacher that predicts the highest confidence in the correct prediction. The prediction distillation loss $L_{KL}$ is defined as follow:

$$L_{KL} = KL(\mathcal{F}(\theta_S, d), \mathcal{F}(\theta_{best}, d)), \theta_{best} = \arg\max_{\theta_i}(\mathcal{F}(\theta_i, d)_j) \tag{7}$$

where $KL(\cdot)$ denotes the KL divergence loss. The overall amalgamation loss $L_{amlg}$ becomes, $L_{amlg} = \lambda_{feat}L_{feat} + \lambda_{rec}L_{rec} + \lambda_{KL}L_{KL}$, where $\lambda$ are used for balancing the loss terms. After knowledge amalgamation, the server model $\theta^*$ is then distributed to each client for extracting features and building prototypes of each category.

While the theoretical complexity of knowledge amalgamation is $O(n)$, practical considerations arise as the number of clients $n$ increases, leading to an increase in the number of FPMs that require training. This poses a challenge due to the limited memory capacity of GPUs, as training all FPMs simultaneously may not be feasible. To address this issue, we suggest a hierarchical amalgamation framework that alleviates the high GPU memory requirement. By allocating a GPU memory budget, we can amalgamate groups of $G$ teacher models instead of all $n$ teacher models at once. Although this modification increases the complexity of knowledge amalgamation to $O(n \log n)$, it enables the implementation of knowledge amalgamation on any GPU device capable of simultaneously amalgamating at least two teacher models.

## 4.2 Prototype-based Model Evaluation

The concept of utilizing prototypes in classification models has its roots in the Prototypical Networks introduced by Snell et al. [47], which were initially proposed for addressing few-shot classification tasks. In this framework, the classification of an embedded query point is achieved by identifying the nearest class prototype. The underlying assumption is that an embedding space exists where data points tend to cluster around a single prototype representation for each class. By leveraging prototypes, it becomes unnecessary to compute the distances between all samples in the query set and the support set, as required in Sung et al. [50]. The prototypes capture the distribution of samples in the support set, simplifying the classification process.

In our study, we adopt a similar perspective to Snell et al. [47] by considering the clients' local datasets as the support set. From the embedding space derived through the Federated Knowledge Amalgamation described earlier, we construct prototypes for each client, denoted as $\mathcal{P}_i^{(j)}$, representing the prototype for class $j$ of the $i$-th client. To address the challenge of evaluation complexity dependence on the size of the validation set, we introduce a modification to the evaluation step. Rather than iterating through all validation samples, we devise a strategy where prototypes are built for the server's validation set. These prototypes, denoted as $\mathcal{P}_\mathcal{V}^{(j)}$, represent the prototypes for class $j$ of the validation set. Model evaluation is then performed by comparing the similarities of the prototypes between the clients and the server. This modification effectively eliminates the computational burden associated with iterating through all validation samples. The prototypes are built as follows:

$$\mathcal{P}_i^{(j)} = \frac{1}{|\mathcal{D}_{i,j}|} \sum_{(x,y) \in \mathcal{D}_{i,j}} f(\theta^*, x) \tag{8}$$

where $\mathcal{D}_{i,j}$ implies the data that is class $j$ in the client $i$-th dataset and function $f(\cdot)$ is used to extract feature from data $x$ with model weight $\theta^*$. We set the prototype of clients as a zero vector if the client does not possess any data of the corresponding class, such that $\mathcal{P}_i^{(j)} = \mathbf{0}$ if $|\mathcal{D}_{i,j}| = 0$. When clients cooperate in coalitions, the prototypes for class $j$ of coalition $\mathcal{S}$ is denoted as $\mathcal{P}_\mathcal{S}^{(j)}$, which can simply be achieved by the weighted sum of prototypes in the coalition.

$$\mathcal{P}_\mathcal{S}^{(j)} = \sum_{i \in \mathcal{S}} \frac{|\mathcal{D}_{i,j}|}{\sum_{i \in \mathcal{S}} |\mathcal{D}_{i,j}|} \mathcal{P}_i^{(j)} \tag{9}$$

We then define the performance of the coalition by the confidence of correctly classifying the given prototypes of the validation set. The confidence is calculated by the relative value between the

similarity of the same class and the sum of the similarities of all classes, while a softmax function is applied to ensure the value lies within the range $[0, 1]$. To be more specific, the value is computed as follows:

$$V'(\mathcal{S}) = \frac{1}{|\mathcal{C}|} \sum_j \frac{e^{Sim(\mathcal{P}_\mathcal{V}^{(j)}, \mathcal{P}_\mathcal{S}^{(j)})}}{\sum_k e^{Sim(\mathcal{P}_\mathcal{V}^{(j)}, \mathcal{P}_\mathcal{S}^{(k)})}}, \tag{10}$$

where $|\mathcal{C}|$ represents the number of classes in the classification task. We adopt cosine similarity as our similarity function.

### 4.3 Contribution Evaluation

Shapley value is applied to evaluate the participant contribution. However, adopting the model performance $V'$ as a utility function may usually lead to a violation of rationality. This violation occurs when the marginal performance gain from additional clients decreases as the coalition size increases, as observed in [32]. Consequently, the expected value of the marginal gain becomes smaller than the individual value, thereby violating individual rationality [3], which states that $c_i \geq u(\{i\})$, where $u(\cdot)$ denotes a utility function. To illustrate this issue, we provide an example below:

**Example 4.1.** Given two clients, each client's local dataset allows them to achieve model accuracy of $0.8$ and $0.6$, respectively. However, when collaborating in a federated learning setting, they can attain a model performance of $0.9$. The Shapley value of each participant can be determined by using the model performance as the utility function, resulting in Shapley values of them becoming $0.55$ and $0.35$. Thus, their payoffs are lower with cooperation.

Despite the seemingly irrational nature of the decision, data owners in real-world scenarios demonstrate a willingness to participate in the federation in the above example due to the higher level of user satisfaction achieved by a model with an accuracy of $0.9$. Relying solely on model performance as the utility function fails to adequately capture the underlying user satisfaction. Based on this observation, we rectify the conventional utility function with a logistic function [9], representing the percentage of users satisfied with the model performance.

$$V(\mathcal{S}) = \frac{1}{1 + e^{-k(V'(S)-T)}} \tag{11}$$

where $k$ denotes the logistic growth rate, which determines the steepness of the curve and $T$ represents the threshold, where half of the population is satisfied. The choice of $T$ should be greater than $\max_i V'(\{i\})$ to encourage the cooperation of participants because the convexity of the left-sided logistic function would up-weight the utility value of stronger coalitions. When $k$ is set to infinity, the individual rationality would be satisfied since the $V(\{i\})$ is compressed to zero and participant contribution is lower-bounded by zero. [1]

Finally, the contribution of $i$-th participant can be calculated by Shapley value defined in Equation 4. The proposed rectification of the utility function with the logistic function makes the utility value more sparse because the utility of the coalition whose performance does not surpass the threshold $T$ would be highly compressed. Inspired by [32], we propose a pruning technique that leverages the sparsity to further accelerate the computation of the Shapley value. We set a pruning threshold $\tau$, such that given a coalition $\mathcal{S}$ with $V(\mathcal{S}) \leq \tau$, we prune the computation of subsets of $\mathcal{S}$. [2]

### 4.4 Complexity Analysis

Table 1 presents a comparative analysis of SPACE's computational complexity in contrast to established methods. The sampling-based approximation of the Shapley value, applied in TMC-Shapley, GTG-Shapley, and SPACE, necessitates sampling complexity of $n \log n$, as studied in [35]. It is worth noting that SPACE achieves a notably reduced computational complexity in calculating the Shapley value at merely $O(n \log n \cdot |\mathcal{C}|)$, which represents a substantial reduction compared to the $O(n \log n \cdot R_g \cdot |\mathcal{D}_\mathcal{V}|)$ required by other Shapley value-based approaches. This efficiency gain is attributed to the use of the single-round amalgamation and the prototype-based model evaluation. The prototype-based model evaluation eliminates the necessity for repeated iteration over the validation set. Consequently, the computation of the Shapley value is facilitated by direct comparisons of

---

[1]Further discussion is provided in the appendices.

[2]The overall algorithm of SPACE is shown in appendices.

Table 1: Comparison of complexity between previous works. $R_l$, $R_g$, and $R_a$ denote the number of iterations for local training, global communication rounds, and knowledge amalgamation. $|\theta|$ represents the number of model parameters, $|\mathcal{P}|$ denotes the dimension of prototypes, and $|\mathcal{C}|$ indicates the number of classes. $|\mathcal{D}_i|$ and $|\mathcal{D}_\mathcal{V}|$ represent the sizes of the local dataset and the validation set.

| Method | Communication Cost | Client Computation Cost | Server Computation Cost |
|---|---|---|---|
| Real Shapley | $2^n \cdot R_g \cdot |\theta|$ | $2^n \cdot R_g \cdot R_l \cdot |\mathcal{D}_i|$ | $2^n \cdot |\mathcal{D}_\mathcal{V}|$ |
| GT-Shapley | $n(\log n)^2 \cdot R_g \cdot |\theta|$ | $n(\log n)^2 \cdot R_g \cdot R_l \cdot |\mathcal{D}_i|$ | $n(\log n)^2 \cdot |\mathcal{D}_\mathcal{V}|$ |
| TMC-Shapley | $n \log n \cdot R_g \cdot |\theta|$ | $n \log n \cdot R_g \cdot R_l \cdot |\mathcal{D}_i|$ | $n \log n \cdot |\mathcal{D}_\mathcal{V}|$ |
| GTG-Shapley | $R_g \cdot |\theta|$ | $R_g \cdot R_l \cdot |\mathcal{D}_i|$ | $n \log n \cdot R_g \cdot |\mathcal{D}_\mathcal{V}|$ |
| DIG-FL | $R_g \cdot |\theta|$ | $R_g \cdot R_l \cdot |\mathcal{D}_i|$ | $\boldsymbol{R_g \cdot |\mathcal{D}_\mathcal{V}|}$ |
| SPACE(Ours) | $\boldsymbol{|\theta| + |\mathcal{P}| \cdot |\mathcal{C}|}$ | $\boldsymbol{R_l \cdot |\mathcal{D}_i|}$ | $n \log n \cdot R_a \cdot |\mathcal{D}_\mathcal{V}|$ |

distances between the weighted sum of prototypes. Moreover, by employing the knowledge amalgamation technique, SPACE is capable of evaluating participant contributions with a single communication round, thereby significantly alleviating the communication and computational load on clients with limited resources by a factor of $R_g$. This acceleration does introduce an additional computation load for knowledge amalgamation on the server, with a complexity of $O(n \log n \cdot R_a \cdot |\mathcal{D}_\mathcal{V}|)$. However, this trade-off is generally more suitable for generic federated learning frameworks, given that the server typically possesses superior computational resources compared to the clients.

# 5 Experiment

In this section, we demonstrate the effectiveness of the proposed SPACE method for evaluating participant contributions. We compare it with previous approaches and show that SPACE achieves superior computational efficiency without compromising performance. We also evaluate the SPACE on two federated-learning tasks, client reweighting, and client selection.

## 5.1 Experimental Setup

**Datasets and Non-IID Setting.** Following [57], we conduct experiments on the widely adopted image dataset *MNIST* [24] and *CIFAR10* [22]. For a fair comparison, we adopted the experimental settings used in the HFL framework [57] with two different scenarios: mislabeled data and non-IID data. In the mislabeled data scenario, we deliberately introduced label corruption in a variable proportion of the total clients. Specifically, we corrupted different percentages of clients, ranging from 0% to 80%. Within each client, we intentionally corrupted 50% of the local training labels to simulate the presence of mislabeled data. For the non-IID scenario, we selected a subset of clients, ranging from 0% to 80%, and assigned them local datasets containing incomplete categories. We set the total number of clients to 10 for the MNIST dataset and 5 for the CIFAR10 dataset.

**Baselines and Implementation.** We compare SPACE with four state-of-the are methods, including *i) GT-Shapley* [19], *ii) TMC-Shapley* [16], *iii) GTG-Shapley* [32] and *iv) DIG-FL* [57]. [3] We also compare a variant of SPACE, *v) SPACE(Avg)*, which applies FedAvg instead of knowledge amalgamation for aggregation. For adjusting the utility function, we empirically set $k$ as 100 while $T$ as 0.95 and 0.5 for evaluation on MNIST and CIFAR10.

**Evaluation Metric.** The performance of all approaches is assessed from two perspectives: accuracy and efficiency. To evaluate the accuracy, we employ Pearson's Correlation Coefficient to measure the correlation between the estimated and actual Shapley values. The actual Shapley values are obtained by performing retraining for a total of $2^n$ times. In terms of efficiency, we consider two vital metrics. Execution time (in seconds) is measured on a single V100 GPU without parallel training to assess the time efficiency of the approaches. Communication cost (in megabytes) is calculated according to the formula $Comm = 2 \cdot n \cdot R_g \cdot |\theta|$, where $R_g$ denotes the number of communication rounds, $|\theta|$ represents the number of model parameters(MB) and constant of 2 stands for the upload and download of the models.

---

[3] Since GT-Shapley and TMC-Shapley are centralized learning schemes, we set the retraining rounds for TMC-Shapley to $n \log n$, and for GT-Shapley to $n(\log n)^2$, where $n$ is the number of participants for the federated learning scenario.

Table 2: Results of participant contribution evaluation.

| Dataset | Scenario | GT | TMC | GTG | DIG-FL | SPACE(Avg) | SPACE |
|---------|----------|-----|------|-----|--------|------------|-------|
| MNIST | Non-IID | 0.6877 | **0.9824** | 0.9287 | 0.8715 | 0.9713 | 0.9448 |
| | Mislabel | 0.4230 | 0.9507 | 0.9608 | 0.9580 | 0.9529 | **0.9612** |
| | Time(s) | 97137 | 84796 | 62473 | 315 | 294 | **160** |
| | Comm(MB) | 11813.12 | 10292.48 | 35.2 | 35.2 | 35.2 | **1.76** |
| CIFAR | Non-IID | 0.6089 | 0.8877 | 0.8208 | 0.7546 | **0.9540** | 0.9290 |
| | Mislabel | 0.5192 | 0.9595 | 0.4148 | 0.9598 | 0.9565 | **0.9641** |
| | Time(s) | 7468 | 4950 | 835 | 536 | 315 | **295** |
| | Comm(MB) | 307800 | 202920 | 11400 | 11400 | 11400 | **570** |

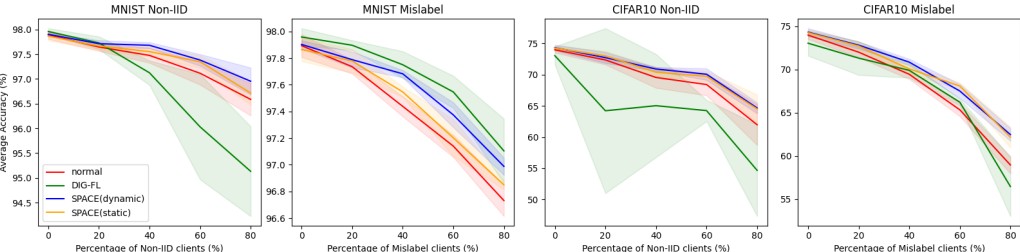

Figure 2: The effect of client reweighting on model performance.

## 5.2 Quantitative Results

**Participant contribution.** Table 2 presents the outcomes of participant contribution evaluation. The proposed SPACE approach demonstrates outstanding efficiency and achieves the highest Pearson correlation coefficient (PCC) in scenarios involving mislabeled data. In terms of PCC, SPACE performs similarly to SPACE(Avg), but with superior efficiency resulting from the single-round knowledge amalgamation. The communication and computational costs associated with GT-Shapley and TMC-Shapley are excessively high due to the requirement for model retraining. However, TMC-Shapley shows improved efficiency by attaining high PCC results with fewer sampled permutations. GTG-Shapley adopts a reconstruction technique to expedite the computation process, which effectively reduces the required communication cost, yet the testing time in GTG-Shapley remains considerably long. The resource-saving version of DIG-FL achieves comparable computational efficiency to SPACE. Nevertheless, it has a tendency to overestimate the contribution of clients with incomplete categories and exhibits inferior PCC results in non-IID scenarios.

**Client Reweighting.** In non-IID scenarios or when some clients have erroneous data, aggregating local models based on data ratio may lead to suboptimal performance. Leveraging participant contribution as a weighting factor can enhance the robustness of federated learning [57]. We propose two reweighting mechanisms, namely static and dynamic, with the former employing participant contribution calculated using the knowledge amalgamated model and the latter recalculating participant contribution for each communication round using the current model. Figure 2 shows the reweighting results. Despite the dynamic approach introducing additional computational overhead, it yields improved performance. The static and dynamic approaches achieve accuracy improvements of 2.53% and 2.69%, respectively, under the non-IID scenario, and 3.21% and 3.52%, respectively, under the mislabel scenario, for the CIFAR10 dataset. The DIG-FL approach measures client contribution in federated learning by the similarities between the gradients of the local datasets and the gradients of the validation set. However, this approach disregards clients in communication rounds where the two gradients diverge, leading to significant variances in the federated training process.

**Client Selection.** Client selection is a critical issue when dealing with a large number of clients. Li et al. [28] proposed a client sampling approach based on the multinomial distribution (MD) that ensures zero bias but has significant variances. To tackle this issue, Fraboni et al. [13] suggested clustered sampling (CS), which has been theoretically shown to reduce variance. They proposed two CS algorithms: Algorithm 1 clusters clients based on data ratio, and Algorithm 2 incorporates the similarities between local and global models. Algorithm 2 outperforms both MD and Algorithm 1. However, Algorithm 2's time complexity is $O(R_g \cdot (n^2|\theta| + X))$, where $|\theta|$ is the number of model parameters and $O(X)$ is the complexity of the clustering algorithm. This computational cost

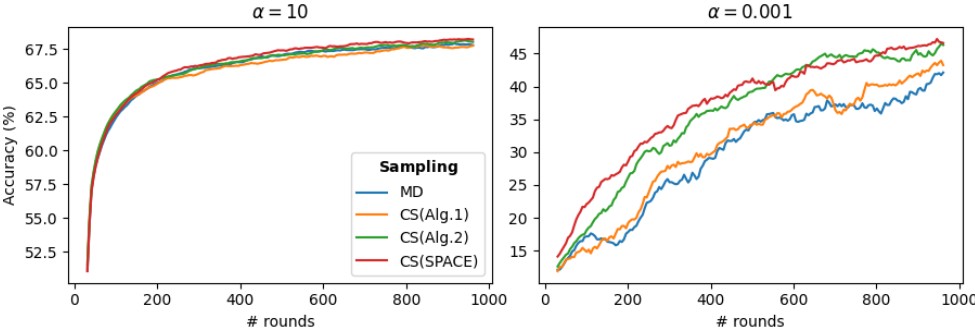

Figure 3: The effect of client selection on model performance.

results from updating a similarity matrix and re-clustering at each communication round. We propose a method that combines SPACE with CS. We utilize the prototypes obtained from SPACE to construct the similarity matrix, eliminating the need for repeated updates. Additionally, we enhance the weighting by incorporating the estimated contribution, resulting in a new weighting scheme for client $p_i$ given by $(1 - \beta)p_i + \beta c_i$, where $\beta$ is a hyper-parameter that balances the two terms. As in [13], the CIFAR10 dataset is distributed among 100 clients using a Dirichlet distribution with $\alpha \in \{10, 0.001\}$, where smaller $\alpha$ implies larger heterogeneity. We set $\beta = 0$ and $\beta = 0.5$ when $\alpha = 10$ and $\alpha = 0.001$, respectively. Figure 3 presents the results of client selection. Our approach, which combines prototypes and participant contribution, outperforms prior methods, indicating the superiority of our proposed method. This improvement can be attributed to the fact that prototypes enhance the accuracy of the clustering process. Furthermore, as discussed in Section 5.2, the incorporation of participant contribution has proven to be useful in enhancing the performance of federated learning, especially in non-IID settings.

## 6 Conclusion

We propose Single-round Participant Amalgamation for Contribution Evaluation (SPACE), an efficient approach for evaluating participant contribution in federated learning. The novelty of SPACE lies in two key components: Federated Knowledge Amalgamation and Prototype-based Model Evaluation. By leveraging a single communication round, Federated Knowledge Amalgamation constructs a robust embedding space, while Prototype-based Model Evaluation reduces the complexity associated with the validation set size. Our experimental results demonstrate that SPACE outperforms existing methods in contribution evaluation and shows versatility in client reweighting and selection. However, efficient client selection for amalgamation and the impact of the validation set quality are areas that require further exploration in future research.

## Broader Impact

In our study, we highlight the importance of participant contribution through three applications: incentive mechanisms, client reweighting, and client selection. Furthermore, the potential applications extend to client debugging and attack detection. Since the computational complexity associated with participant contribution hampers its widespread implementation, we propose a novel FL framework, SPACE, to alleviate this issue and pave the way for broader adoption of participant contribution. An efficient algorithm for evaluating participants provides decision-makers with valuable insights before initiating extensive federated learning processes, which can be both computationally and temporally intensive. Furthermore, rapid evaluation offers enhanced flexibility when dealing with dynamic cooperative entities, facilitating quick adaptation to new collaborations—a capability that previous methods lack. Our proposed approach, SPACE, significantly contributes to these applications. However, our framework relies on the assumption of having a proper and reliable validation set with accurately labeled data. This prerequisite can be particularly challenging to fulfill in real-world settings and thus limits the application to specific scenarios characterized by significant commercial incentives [33, 65]. Exploring methods to decrease dependence on the validation set while maintaining functionality, even when erroneous data impact many clients, presents an intriguing direction for future exploration. We defer this critical aspect to be addressed in future research efforts.

## Acknowledgement

This work was supported in part by the National Science and Technology Council, Taiwan, under grant NSTC-112-2223-E-002-015, and by the Ministry of Education, Taiwan, under grant MOE 112L9009.

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

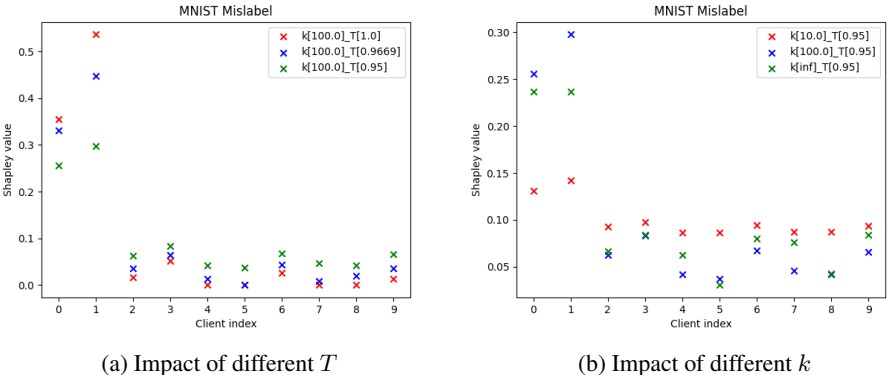

(a) Impact of different $T$            (b) Impact of different $k$

Figure 4: The impact of modification using the logistic function with different parameters.

# A   Additional Experimental Results

## A.1   Sensitivity test of utility function

To begin with, we conduct a sensitivity analysis on the utility function by varying two hyperparameters: $k$ and $T$. The hyperparameter $k$ determines the logistic growth rate, thereby impacting the steepness of the curve, while $T$ represents the threshold at which half of the population is considered satisfied. This adjustment is aimed at capturing user satisfaction, which serves as a practical incentive for participants to actively participate in federated learning (FL) collaborations.

Figure 4 depicts the calculated actual Shapley values for the MNIST dataset in the presence of mislabeled data, considering the existence of 8 corrupted clients. Various combinations of the hyperparameters $k$ and $T$ are explored in this analysis. Specifically, Figure 4a focuses on investigating the impact of different values of $T$ from the set $1.0, 0.9669, 0.95$. The selection of $1.0$ as the threshold follows a greedy approach, while $0.9669$ represents the performance achieved when considering all participants. Furthermore, $0.95$ corresponds to an empirically derived threshold based on a comprehensive dataset understanding.

The findings reveal that selecting a greedy selection for $T$ leads to the disparity between clients with high and low contributions, due to the convex nature of the left-sided logistic function. Conversely, empirically setting an appropriate value for $T$ reduces the marginal gain for coalitions that surpasses the threshold, owing to the concave property of the right-sided logistic function. Consequently, this approach minimizes the discrepancy between clients with high and low contributions.

Figure 4b examines the influence of different values of $k$. A larger value of $k$ indicates a steeper logistic function, thereby strongly favoring coalitions that surpass the threshold $T$ and consequently widening the gap between clients with high and low contributions. However, as $k$ approaches infinity, the logistic function transforms into a ReLU function, disregarding all coalitions that fail to surpass the threshold $T$ and assigning equal value to those that do. Consequently, in this scenario, the relative performance, as depicted by the difference between client 0 and client 1 in Figure 4b, would be disregarded due to a hard threshold.

The selection of hyperparameters $k$ and $T$ should be determined by the desired characteristics of the applications to enhance the influence of coalitions with superior performance or to ensure individual rationality. Thus, we maintain them as carefully selected hyperparameters, allowing for flexibility and customization based on specific application requirements.

## A.2   Ablation Study for smaller $G$

The federated knowledge amalgamation process utilizes FPMs comprising only two convolution layers, each of which introduces minimal additional GPU memory usage. As a result, in the reported experiments, we are able to simultaneously distill the knowledge from all client models on a single V100 GPU. However, as the number of clients $n$ increases, the limited memory capacity of the GPU

Table 3: Federated knowledge amalgamation with different $G$.

| | MNIST | | | CIFAR10 | | |
|---|---|---|---|---|---|---|
| G | Non-IID | Mislabel | runtime | Non-IID | Mislabel | runtime |
| $n$ | 0.9448 | 0.9612 | 160 | 0.9290 | 0.9641 | 295 |
| 2 | 0.9427 | 0.9726 | 633 | 0.9333 | 0.9645 | 299 |

Table 4: Results of participant contribution evaluation on Tiny-ImageNet dataset.

| Dataset | Scenario | GT | TMC | GTG | DIG-FL | SPACE(Avg) | SPACE |
|---|---|---|---|---|---|---|---|
| Tiny-ImageNet | Non-IID | 0.9082 | **0.9610** | 0.7425 | 0.8944 | 0.9175 | 0.9092 |
| | Mislabel | 0.7833 | 0.9233 | 0.7998 | 0.8405 | **0.9293** | 0.9256 |
| | Time(s) | 84994 | 69212 | 98855 | 266 | **252** | 453 |
| | Comm(MB) | 2639200 | 2097600 | 8000 | 8000 | 8000 | **400** |

may pose a challenge. To address this issue, we propose a hierarchical amalgamation framework that mitigates the high GPU memory requirement.

To evaluate the effectiveness of hierarchical amalgamation, we conducted experiments by amalgamating groups of $G = 2$ teacher models instead of all $n$ teacher models in a hierarchical manner. The results, shown in Table 3, demonstrate that knowledge transfer from the local datasets can still be achieved through hierarchical amalgamation, yielding performance comparable to amalgamating all clients simultaneously. Although hierarchical amalgamation alleviates the high GPU memory requirement, it introduces a computational complexity increase in the knowledge amalgamation process from $O(n)$ to $O(n \log n)$, as evident from Table 3.

### A.3 Comparison of estimated Shapley Value

Figure 5 illustrates the scatter plots depicting the estimated Shapley values compared to the actual Shapley values. Regarding the centralized method, TMC-Shapley outperforms GT-Shapley, indicating that Monte Carlo sampling is more efficient for approximating the Shapley value than group testing. To accelerate the approximation process, GTG-Shapley employs gradient-based model reconstruction. While it performs well on the MNIST dataset, its performance deteriorates on the CIFAR10 dataset due to the increased complexity of both the model and the dataset. The results indicate that DIG-FL tends to overestimate clients' contributions with incomplete class data, leading to suboptimal performance in non-IID scenarios. Conversely, the proposed SPACE approach exhibits strong performance across all scenarios.

### A.4 Results on Tiny-ImageNet

To assess the efficacy of the proposed SPACE framework on more complicated datasets, an additional experiment was conducted, incorporating a portion of the Tiny-ImageNet dataset [23]. This subset encompasses 50 categories and involves 10 clients. The results are provided in Table 4. The SPACE method demonstrates satisfying performance in terms of Pearson's Correlation Coefficient (PCC). It's noteworthy that achieving feature alignment in a complex dataset like Tiny-ImageNet is more challenging. Consequently, the knowledge amalgamation process requires an extended number of epochs to converge, particularly in scenarios where the data distribution across clients is non-IID, and clients possess data from disjoint classes. On the other hand, the SPACE(Avg) approach, which employs FedAvg for obtaining the embedding space, may offer superior time efficiency in cases with a limited number of communication rounds. Nonetheless, the advantages in communication efficiency stemming from single-round amalgamation continue to position SPACE as the preferred choice when mitigating communication costs is a key consideration.

### A.5 Scenario without labeled validation set

Given the inherent challenges associated with acquiring a sufficiently well-labeled validation dataset, there arises an important question regarding the adaptability of the proposed approach in cases where label information is unavailable. To investigate this, we conducted supplementary experiments employing unlabeled data for knowledge amalgamation. In this context, we drew inspiration from the work of [44] and modified the loss function to incorporate weights based on entropy purity rather than the confidence associated with accurate result predictions. Specifically, clients with lower en-

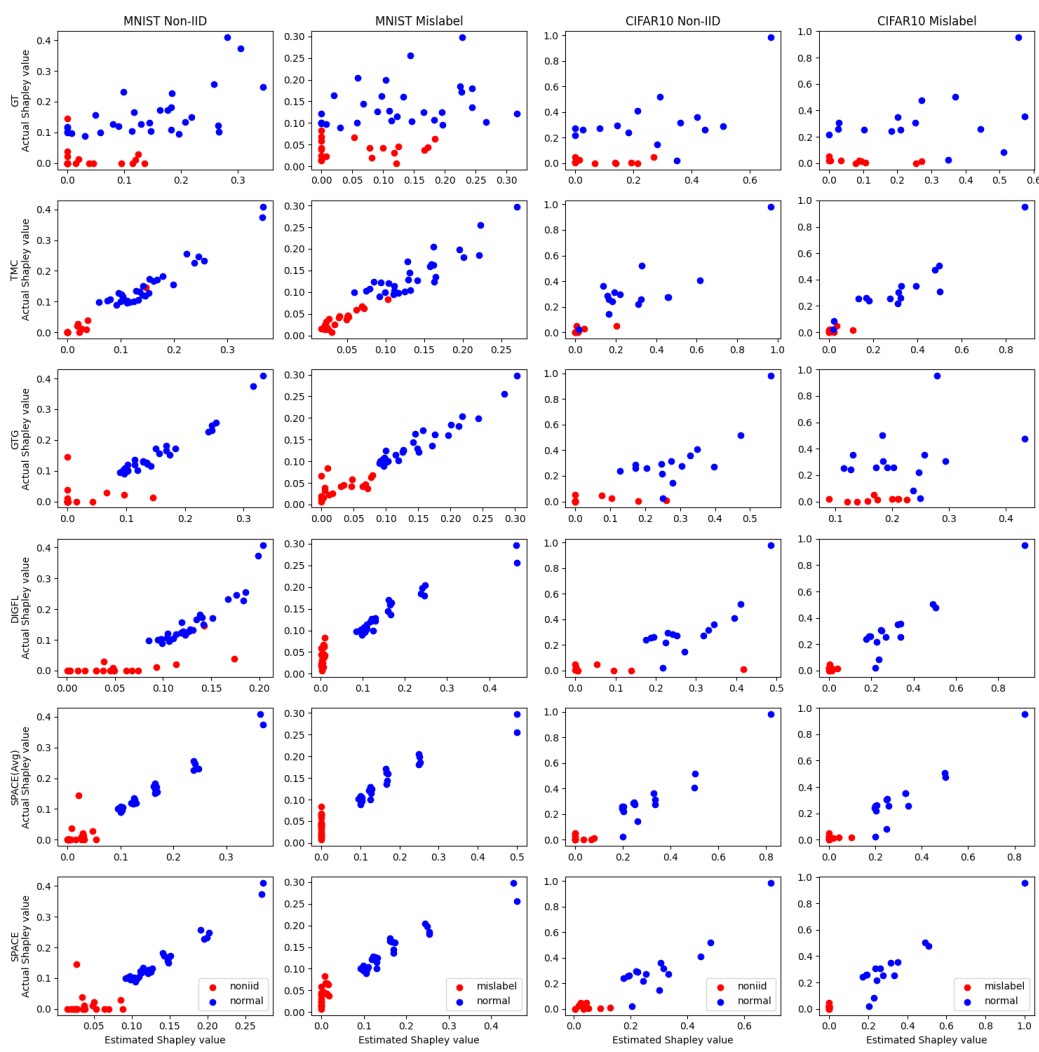

Figure 5: Estimated Shapley and the actual Shapley value of all approaches under all scenarios.

Table 5: Results of SPACE with and without labels.

| | MNIST | | CIFAR10 | | Tiny-ImageNet | |
|---|---|---|---|---|---|---|
| Label | Non-IID | Mislabel | Non-IID | Mislabel | Non-IID | Mislabel |
| ✓ | 0.9448 | 0.9612 | 0.9290 | 0.9641 | 0.9092 | 0.9256 |
| | 0.8772 | 0.9611 | 0.8653 | 0.9641 | 0.9091 | 0.9231 |

tropy values were assigned greater weight in the loss function. The experimental results are shown in Table 5. While some marginal reductions in performance are evident, particularly in scenarios characterized by Non-IID data distributions, it is important to note that these performance decreases are primarily attributed to the absence of label information. Nevertheless, the results clearly demonstrate the viability of leveraging unlabeled data as a practical alternative for knowledge amalgamation, offering a feasible solution in cases where the acquisition of labeled data is challenging.

## B  Pseudocode

The algorithmic representation of the proposed SPACE approach is presented in Algorithm 1. Initially, the server distributes the initial model $\theta$ to all participating clients. Each client then performs empirical risk minimization to obtain a locally optimized model $\theta_i$ and transmits it to the server. Using the locally optimized models as teacher models, the federated knowledge amalgamation tech-

---

**Algorithm 1** SPACE

---

**Require:** Initial FL model $\theta$, client set $\mathcal{N}$, local datasets $\{\mathcal{D}_1, ..., \mathcal{D}_n\}$, validation dataset $\mathcal{D}_\mathcal{V}$, feature encoder $f(\cdot)$ and utility function $V(\cdot)$
  1: # *Knowledge Amalgamation*
  2: **for** $i = 1,\ldots,$n **do**
  3:    $\theta_i = \text{LocalUpdate}(\theta, \mathcal{D}_i)$
  4: $\theta^* = \text{KA}(\theta, \{\theta_1, \ldots, \theta_n\}, \mathcal{D}_\mathcal{V})$
  5: # *Build server prototypes and clients' prototypes*
  6: **for** $j = 1,\ldots,|\mathcal{C}|$ **do**
  7:    $\mathcal{P}_\mathcal{V}^{(j)} = \frac{1}{|\mathcal{D}_{\mathcal{V},j}|} \sum_x f(\theta^*, x), \forall x \in \mathcal{D}_{\mathcal{V},j}$
  8: **for** $i = 1,\ldots,$n **do**
  9:    **for** $j = 1,\ldots,|\mathcal{C}|$ **do**
10:      $\mathcal{P}_i^{(j)} = \frac{1}{|\mathcal{D}_{i,j}|} \sum_x f(\theta^*, x), \forall x \in \mathcal{D}_{i,j}$
11: # *Evaluate contribution with Prototype-based Model Evaluation*
12: r = 0
13: $\{c_1, ..., c_n\} = \mathbf{0}$
14: **while** Convergence criteria not met **do**
15:    r = r + 1
16:    $\pi^r \sim \Pi$
17:    $v_0 = V(\mathcal{N})$
18:    **for** $k = 1,\ldots,$n-1 **do**
19:      **if** $v_{k-1} > \tau$ **then**
20:       $\mathcal{S} = \{\pi^r[k+1], \ldots, \pi^r[n]\}$
21:       **for** $j = 1,\ldots,|\mathcal{C}|$ **do**
22:        $\mathcal{P}_\mathcal{S}^{(j)} = \sum_{i \in \mathcal{S}} \frac{|\mathcal{D}_{i,j}|}{\sum_{i \in \mathcal{S}} |\mathcal{D}_{i,j}|} \mathcal{P}_i^{(j)}$
23:       $v_k = V(\mathcal{S})$
24:      **else**
25:       $v_k = 0$
26:      $c_{\pi^r[k]} = \frac{r-1}{r} c_{\pi^r[k]} + \frac{1}{r}(v_{k-1} - v_k)$
27: **return** $\{c_i, \ldots, c_n\}$

---

nique is employed to distill knowledge into the server model, resulting in an enhanced server model denoted as $\theta^*$. Subsequently, prototypes are constructed using $\theta^*$ for both the validation set and local datasets. Following prototype construction, SPACE proceeds to calculate participant contributions using the Shapley value, leveraging the prototype-based model evaluation. To ensure convergence, SPACE samples multiple permutations of participants. As the modified utility function may compress the utility values of small coalitions, SPACE performs pruning operations at the sequence level. If the utility value $V(\mathcal{S})$ of a coalition $\mathcal{S}$ falls below the pruning threshold, subsets of $\mathcal{S}$ are pruned as their marginal gains become negligible.

## C  Potential Malicious Attack

**Malicious attack by data poisoning.** The manipulation of data labels, a frequently employed malicious strategy by clients engaged in data poisoning attacks, is denoted as "label-flipping" in reference [6]. In our work, we adopted the term "mislabeled" to refer to unintentional and intentional mislabeling instances. In this case, malicious clients attempt to disrupt the training process by submitting gradients derived from mislabeled data. Explicitly, as depicted in Table 2, the results highlight the effectiveness of SPACE in pinpointing those malevolent clients harboring inaccurately labeled data. With the server's privately owned validation set, our framework efficiently identifies such malicious clients, even if they form the majority in the federated process. In addition, the experimental results in Figure 2 demonstrate the robustness of our framework in the federated training. It is worth noting that the SPACE proposed employs contributions as weighting factors, thereby reducing the negative impact of malicious participants, i.e., the malicious clients usually contribute less due to their distributions differing from that of the server.

**Maliciously inflating contribution.** Clients may aim to inflate their contributions maliciously to obtain more significant rewards. If the prototypes from the validation set become exposed, clients might exploit this knowledge to boost their contributions in an ad-hoc manner. In such a case, the SPACE utilizes prototype-based model evaluations, where all clients have access solely to the feature extraction layers of the amalgamated model while excluding access to the fully-connected layers, i.e., the final score. This strategic approach facilitates the creation of prototypes without compromising the confidentiality of the distribution that underlies the validation set. Note that another potential threat is clients employing the zeroth-order optimization technique [4] to derive prototypes with disproportionately high contributions. In this case, we can simply detect this attack by limiting the client's access to the server in that the SPACE does not require each client to recalculate their contributions frequently.

In addition to the aforementioned attacks directly associated with contribution evaluation, federated learning is known for its vulnerability to a spectrum of attacks [6], including but not limited to reconstruction attacks [74]. It is noted that to counter these threats, researchers have proposed defense mechanisms, such as gradient pruning [74], data representation perturbation [49], and differential privacy (DP) [60]. These works are essential. However, they are, in essence, orthogonal to the main theme of this paper, which focuses on contribution evaluation. Nevertheless, these privacy-preserving strategies can be integrated with our method, working collaboratively to enhance data privacy protection.

