# OpenReview forum: "SPACE: Single-round Participant Amalgamation for Contribution Evaluation in Federated Learning"
_NeurIPS.cc/2023/Conference — NeurIPS 2023 poster_

### Official Review · Reviewer_HEwC · 2023-07-04

**Soundness:** 3 good
**Presentation:** 3 good
**Contribution:** 3 good
**Rating:** 6
**Confidence:** 4

**Summary:**

This paper introduced a novel method to evaluate the participant contribution under the setting of federated learning. In the beginning, the author raised two challenges of recent works related to the participant contribution of FL: Multi-round training and dataset dependency, which are actually about the communication and computational costs in FL. The author derived the SPACE model to address these two challenges. Specifically, to address the high communication cost issue, the author introduced a module named Federated Knowledge Amalgamation, which enables a single round of communication. Then the author introduced Prototype-based Model Evaluation to reduce the evaluation complexity since this method only needs similarities comparison without iterating through a validation set. In addition, in the contribution part, the author rectified the utility function with a logistic function to address the rationality violation issue. Last, the author evaluated participant contributions to the SPACE model on two datasets compared with four baselines. Furthermore, the author experimented with SPACE on the client reweighting and selection problem.

**Strengths:**

1. The author raised two challenges of participant contribution evaluation in FL, and introduces corresponding techniques to address them.
2. The author explained the reason and function of each module in detail.
3. The author provides a theoretical value of communication costs and computational costs of baselines and derived model.


**Weaknesses:**

1. The author should pay attention to the writing style. For example, more focus on key points, like how to reduce computational complexity. Otherwise, it is hard to follow.
2. The author should provide more detail about the theoretical analysis on communication and computational cost. For example, how the design has the advantage to reduce complexity and appropriate for GPU?
3. In section 4.3, the author pointed out a problem that applying model performance as utility function may violate rationality. And the author add a activation as solution. Could you explain why you choose this activation and why this design can avoid violations?
4. The focus in this paper is to reduce complexity, so the author should provide results of the computational cost and communication cost in experiment part.


**Questions:**

Like the suggestion in weakness:
1. The author should provide more detail about the theoretical analysis on communication and computational cost. For example, how the design has the advantage to reduce complexity and appropriate for GPU?
2. In section 4.3, the author pointed out a problem that applying model performance as utility function may violate rationality. And the author add a activation as solution. Could you explain why you choose this activation and why this design can avoid violations?
3. The focus in this paper is to reduce complexity, so the author should provide results of the computational cost and communication cost in experiment part.


**Limitations:**

The author can provide more details of the theoretical complexity analysis. In addition, a more comprehensive experiments should be conducted, such as including model complexity, ablation study and case study.

---

> ### Author Rebuttal · Authors · 2023-08-09
>
> We thank the reviewer for the positive review and constructive comments. We provide our responses as follows.
> > The author should pay attention to the writing style. For example, more focus on key points, like how to reduce computational complexity. Otherwise, it is hard to follow.
>
> Thanks for pointing out this issue. We have already provided the detailed complexity analysis in the supplementary materials and will move to the main manuscript in the camera-ready version if space allows.
>
> > The author should provide more detail about the theoretical analysis on communication and computational cost. For example, how the design has the advantage to reduce complexity and appropriate for GPU?
>
> Thanks. We have implemented the hierarchy amalgamation technique to infer SPACE to minimize overall GPU memory usage. Specifically, as the number of clients 'n' increases, the number of Feature Projector Modules (FPMs) that necessitate training also escalates. Rather than conducting the amalgamation of all clients simultaneously, leading to a substantial demand for peak GPU memory, our proposed hierarchy amalgamation approach facilitates the amalgamation of fewer clients at a time, effectively reducing the peak GPU memory requirements.
>
> > In section 4.3, the author pointed out a problem that applying model performance as utility function may violate rationality. And the author add a activation as solution. Could you explain why you choose this activation and why this design can avoid violations?
>
> We employ the sigmoid function because of its capacity to represent the characteristics of user satisfaction aptly. As model performance surpasses a certain threshold, the number of satisfied users increases significantly, but eventually, the satisfaction tends to saturate for exceedingly high-performance levels. For instance, a model with a performance of 97% may not exhibit substantially different user satisfaction compared to one with a performance of 98%. The sigmoid function effectively captures these two aspects of user satisfaction. To ensure adherence to individual rationality, we can select a sigmoid function with large values for the parameters $k$ and set $T$ between the greatest individual performance and the full coalition performance. This configuration substantially diminishes the individual value while amplifying the value of coalitions, thus mitigating the risk of violating individual rationality.
>
> > The focus in this paper is to reduce complexity, so the author should provide results of the computational cost and communication cost in experiment part.
>
> Thank you for the valuable suggestion. Indeed, we have already included the computational cost in terms of runtime in Table 2, indicating the computational efficiency of our proposed method. To further illustrate the communicational efficiency, we will supplement the communication cost in terms of the amount of data transmitted, measured in megabytes (MB), as follows. Note that the communication cost shown in the table is calculated as $Comm = 2 \cdot n \cdot R_g \cdot |\theta|$, where $R_g$ denotes the number of communication rounds, $|\theta|$ represents the number of model parameters(MB) and constant of 2 stands for the upload and download of the models. For approaches that require retraining, the number of clients $n$ involves repeated clients during retraining.
>
> The findings concerning communication costs underscore the notable communication efficiency inherent in the proposed SPACE framework. The adoption of a single-round amalgamation strategy results in a substantial decrease in communication costs, yielding a scale advantage proportional to a factor of $R_g$, especially in comparison to approaches that demand the complete training of the federated model.
>
> * Result on MNIST
>
> |           |  GT       | TMC   | GTG  | DIG-FL  | SPACE(Avg)  | SPACE  |
> |---        |---        |---    |---|---|---|---|
> | Non-IID   | 0.6877    |  **0.9824**    | 0.9287   | 0.8715 | 0.9713   | 0.9448    |
> | Mislabel  | 0.4230    |  0.9507    | 0.9608   | 0.9580 | 0.9529   | **0.9612**    |
> | Times(s)  | 97137     |  84796     | 62473    | 315    | 294      | **160**   |
> | Comm(MB)  | 11813.12  |  10292.48  | 35.2     | 35.2   | 35.2     | **1.76**   |
>
>
> * Result on CIFAR
>
> |           |  GT       | TMC   | GTG  | DIG-FL  | SPACE(Avg)  | SPACE  |
> |---        |---        |---    |---|---|---|---|
> | Non-IID   | 0.6089    |  0.8877     | 0.8208  | 0.7546   | **0.9540**   | 0.9290    |
> | Mislabel  | 0.5192    |  0.9595     | 0.4148  | 0.9598   | 0.9565   | **0.9641**    |
> | Times(s)  | 7468     |  4950      | 835   | 536      | 315      | **295**   |
> | Comm(MB)  | 307800   |  202920    | 11400    | 11400     | 11400     | **570**   |

---

### Official Review · Reviewer_AA9d · 2023-07-07

**Soundness:** 2 fair
**Presentation:** 3 good
**Contribution:** 2 fair
**Rating:** 5
**Confidence:** 3

**Summary:**

This paper studies to evaluate the contribution of each client during federated training efficiently. It proposes a framework named SPACE, which trains a student model in the server with one communication round to measure the similarity between local datasets and the validation dataset in server. Finally, extensive experiments are conducted to prove the efficiency of SPACE.

**Strengths:**

- The paper is well written and organized.
- The proposed solution is feasible and efficient.
- Extensive experiments are conducted to evaluate the proposed framework.

**Weaknesses:**

- The assumption of SPACE is too strong. (See Question 1 and 2)
- The proposed contribution estimation framework is decoupled from federated training. (See question 3)
- The dataset and client number in the experiment are too small. (See Question 4)

**Questions:**

1. One important assumption in this paper is the server holds validation dataset, which is rare in reality. What if the validation dataset is hold by distributed clients rather than the central server?
2. In section 1, the paper states "if the server and client distributions are similar, the client dataset proves helpful in correctly classifying the data in the validation set of the server."
    - First, just like Equation 1 the target of FL is to cooperatively train a model that performs well on the test datasets in all clients rather than the validation set of the server.
    - Secondly, it seems like the authors assume that the validation dataset in the server shares the same distribution with the distributed test datasets. In my opinion, the assumption is too strong and unpractical. If the server already knows the distribution of the test datasets, there is no need to federated-train a model.
3. The contribution estimation in SPACE is decoupled from federated training. I' curious how does SPACE deal with client over-sampling, where the number of participation for different clients is unbalanced and some clients may be not selected during training.
4. In the experiment, the dataset and the number of clients are too small (5 for CIFAR10 and 10 for MNIST).

**Limitations:**

yes

---

> ### Author Rebuttal · Authors · 2023-08-09
>
> Thank you for your helpful feedback. We have answered all your concerns. In the following, we respond point by point.
> > One important assumption in this paper is the server holds validation dataset, which is rare in reality. What if the validation dataset is hold by distributed clients rather than the central server?
>
> If distributed clients solely retain their validation dataset, SPACE remains operational, albeit with the need for an extra unlabeled server dataset for knowledge amalgamation. Furthermore, a modification of the distillation loss for unlabeled data, as illustrated in [39], becomes necessary. Alternatively, SPACE(Avg) could serve as a simpler alternative for building feature embedding space if access to an additional unlabeled server dataset is unfeasible. After constructing the embedding space, clients must upload the prototypes of their respective validation sets. The global validation prototypes are then obtained by performing a weighted sum of the local validation prototypes, and the contribution can be calculated using the prototype-based evaluation. However, a potential drawback of this setup is that if mislabeled data exists in any of the local validation sets, the global validation set would also be adversely affected.
>
> We provide additional experimental results for  SPACE with unlabeled data for knowledge integration. The assignment of weights to clients in the loss function is guided by their prediction entropy, with greater weighting for those with lower entropy. Although slight decreases in performance are observed, particularly in Non-IID scenarios, these can be attributed to the absence of label information. Nonetheless, the outcomes manifest the feasibility of leveraging unlabeled data as a viable substitute for knowledge integration, especially in practical situations where obtaining labeled data poses difficulties.
>
>
> * Result of SPACE(unlabel)
>
> |  Scenario       | MNIST   | CIFAR  | Tiny-ImageNet  |
> |---        |---    |---|---|
> | Non-IID   | 0.8772    |  0.8653     | 0.9091  |
> | Mislabel  | 0.9611    |  0.9641     | 0.9231  |
>
> > The authors assume that the validation dataset in the server shares the same distribution with the distributed test datasets. In my opinion, the assumption is too strong and unpractical. If the server already knows the distribution of the test datasets, there is no need to federated-train a model.
>
> Thanks for pointing out this issue. Following previous works [29, 43, 51, 52, 53], we assume that the server initially possesses a validation set. This validation set is pivotal for ensuring equitable and reliable assessment of contributions within the context of federated learning. As deliberated in the "Broader Impact" section, we are fully cognizant of the challenges of procuring a trustworthy validation in real-world settings. Nonetheless, specific scenarios exist wherein servers may be motivated to assemble such a validation set in exchange for substantial commercial benefits. Notably, instances like those discussed in [J, K] underscore this idea of servers having access to validation sets for contribution evaluation in real-world applications. In [J], multiple city-gas companies collaborate to train a hazard identification model. Similarly, [K] presents experiments in the healthcare domain involving the participation of eight esteemed medical institutions in China to construct healthcare decision-support models. Both of these applications necessitate a reliable server-based validation to ensure AI models' accuracy objectives before their deployment in real-world scenarios. Though the size of the validation set might not be large enough for centralized training under the setting, it provides a reliable evaluation of the model performance, especially when the clients hold mislabeled data. In light of such examples, we contend that our underlying assumption of the availability of a validation set on the server holds practical in specific scenarios.
>
> > The contribution estimation in SPACE is decoupled from federated training. I' curious how does SPACE deal with client over-sampling, where the number of participation for different clients is unbalanced and some clients may be not selected during training.
>
> The clustered sampling (CS) method [11] is introduced to tackle over-sampling and reduce performance variance in federated learning. Instead of employing multinomial distribution (MD) for sampling with replacement [25], which has exhibited substantial performance variance, the clustered sampling approach first categorizes clients into distinct clusters. Subsequently, clients are sampled from these clusters, ensuring that clients with unique distributions are more likely to be sampled. As mentioned in Section 5.2 within the context of client selection, we first utilize the SPACE framework to derive the prototypes and contributions of individual clients. The prototypes are then utilized for clustering purposes, while clients' contributions determine the clients' probability. This combined approach effectively addresses the over-sampling issue while enhancing the training outcomes.
>
> > In the experiment, the dataset and the number of clients are too small (5 for CIFAR10 and 10 for MNIST).
>
> Please see "Author Rebuttal by Authors" above.
>
> [J] Yang, C., Liu, J., Sun, H., Li, T., & Li, Z. (2022). WTDP-Shapley: Efficient and Effective Incentive Mechanism in Federated Learning for Intelligent Safety Inspection. IEEE Transactions on Big Data.
>
> [K] Liu, Z., Chen, Y., Zhao, Y., Yu, H., Liu, Y., Bao, R., ... & Yang, Q. (2022, June). Contribution-aware federated learning for smart healthcare. In Proceedings of the AAAI Conference on Artificial Intelligence (Vol. 36, No. 11, pp. 12396-12404).

---

> > ### Comment · Reviewer_AA9d · 2023-08-11
> > **Response**
> >
> > - Thanks for your response. It solves part of my concerns, and I'm glad to adjust my rating.

---

### Official Review · Reviewer_AsTD · 2023-07-08

**Soundness:** 2 fair
**Presentation:** 3 good
**Contribution:** 2 fair
**Rating:** 4
**Confidence:** 4

**Summary:**

This paper proposes a single-round participant contribution evaluation method for FL. The novel and interesting part is using the sample embedding similarity between client data and (server) validation data to indicate contribution, thus avoiding the time-consuming model retraining step.

**Strengths:**

1. Very useful problem
2. Interesting idea with using embedding instead of retraining models for contribution evaluation.

**Weaknesses:**

1. No theoretical analysis of why the embedding similarity can lead to an excellent approximation to the original Shapley-based contribution evaluation with re-training. Since embedding similarity-based contribution evaluation is quite different from original retraining-based contribution evaluation, such a theoretical analysis is a must; only empirical analysis cannot verify that the proposed contribution evaluation can always be good.

2. Experiment is too limited. More datasets, more parties, and malicious behaviors of parties should all be counted to ensure that the contribution obtained by the proposed method is robust.

3. The method seems to only work for classification. How to extend it to regression tasks?

**Questions:**

See weakness.

**Limitations:**

Not enough. More discussions on when the proposed model can work well should be given.

---

> ### Author Rebuttal · Authors · 2023-08-09
>
> Thank you for your helpful feedback. We have answered all your concerns. In the following, we respond point by point.
>
> > No theoretical analysis of why the embedding similarity can lead to an excellent approximation to the original Shapley-based contribution evaluation with re-training.
>
> Please see "Author Rebuttal by Authors" above.
>
> > Experiment is too limited. More datasets, more parties, and malicious behaviors of parties should all be counted to ensure that the contribution obtained by the proposed method is robust.
>
> Please see "Author Rebuttal by Authors" above.
>
> > The method seems to only work for classification. How to extend it to regression tasks?
>
> Thanks. Our methodology is primarily designed for classification tasks, drawing from the works cited in references [29, 43, 51, 52, 53]. It is important to note that federated learning also encompasses the significant domain of regression tasks[H, I]. However, it is evident from our literature review that the discourse surrounding the evaluation of contributions in the context of regression tasks could be more extensive. As such, a notable gap exists in the literature regarding the evaluation of contributions within regression tasks, thus signifying a crucial avenue for future exploration.
>
> > More discussions on when the proposed model can work well should be given.
>
> The experimental results indicate that the proposed SPACE approach is effective, especially in a mislabeled data scenario. This is because prototypes obtained from clients with mislabeled data exhibit substantial deviation from the prototypes of the validation set. Consequently, such deviations are efficiently identified by the prototype-based model evaluation approach employed in the proposed method.
>
>
> [H] Lee, H., Bertozzi, A. L., Kovačević, J., & Chi, Y. (2022, May). Privacy-Preserving Federated Multi-Task Linear Regression: A One-Shot Linear Mixing Approach Inspired By Graph Regularization. In ICASSP 2022-2022 IEEE International Conference on Acoustics, Speech and Signal Processing (ICASSP) (pp. 5947-5951). IEEE.
>
> [I] Su, L., Xu, J., & Yang, P. (2022). Global convergence of federated learning for mixed regression. Advances in Neural Information Processing Systems, 35, 29889-29902.

---

> > ### Comment · Reviewer_AsTD · 2023-08-19
> >
> > I appreciate the authors' response to the theoretical analysis. However, I do not see more experiments on users' potential malicious behaviors. I think that a rigorous analysis (both theoretical and empirical) of malicious behaviors is really important to indicate whether the proposed method is practically useful or not. This is because in practice we cannot assume that participants are (semi-)honest, we need to realize potential risks clearly.

---

> > > ### Author Response · Authors · 2023-08-21
> > >
> > > Thanks very much for the valuable comments. We fully agree with the reviewer that considering the threat from malicious clients in federated learning is very important. As a matter of fact, as detaily described below, we indeed discussed two primary forms of possible harmful actions tied to contribution assessment and elaborated via empirical works on how our SPACE could adequately tackle these concerns. However, in our submitted paper, we referred to “the behavior of malicious users” as “mislabelling.” In the revised version of this work, we shall clearly use the term “the behavior of malicious users” so that our work can be better understood in the right context. Explicitly, the two forms, malicious attack by data poisoning and maliciously inflating contribution, below are what we have addressed.
> > >
> > >
> > > Malicious attack by data poisoning:\
> > >  The manipulation of data labels, a frequently employed malicious strategy by clients engaged in data poisoning attacks, is denoted as "label-flipping" in reference [D]. In our originally submitted paper, we adopted the term "mislabeled" to refer to unintentional and intentional mislabeling instances. In this case, malicious clients attempt to disrupt the training process by submitting gradients derived from mislabeled data. Explicitly, as depicted in Table 2 on page 8, the results highlight the effectiveness of SPACE in pinpointing those malevolent clients harboring inaccurately labeled data. With the server's privately-owned validation set, our framework efficiently identifies such malicious clients, even if they form the majority in the federated process. In addition, the experimental results in Figure 2 on page 8 demonstrate the robustness of our framework in the federated training. It is worth noting that the SPACE proposed employs contributions as weighting factors, thereby reducing the negative impact of malicious participants, i.e., the malicious clients usually contribute less due to their distributions differing from that of the server.
> > >
> > > Maliciously inflating contribution: \
> > > Another concern we addressed arises when clients aim to inflate their contributions maliciously to obtain more significant rewards. If the prototypes from the validation set become exposed, clients might exploit this knowledge to boost their contributions in an ad-hoc manner. In such a case, the SPACE utilizes prototype-based model evaluations, where all clients have access solely to the feature extraction layers of the amalgamated model while excluding access to the fully-connected layers, i.e., the final score. This strategic approach facilitates the creation of prototypes without compromising the confidentiality of the distribution that underlies the validation set. Note that another potential threat is clients employing the zeroth-order optimization technique [C] to derive prototypes with disproportionately high contributions. In this case, we can simply detect this attack by limiting the client's access to the server in that the SPACE does not require each client to recalculate their contributions frequently. As suggested, we will include the above in our revised paper for better clarification.
> > >
> > > In addition to the aforementioned attacks directly associated with contribution evaluation, federated learning is known for its vulnerability to a spectrum of attacks [D], including but not limited to reconstruction attacks [E]. It is noted that to counter these threats, researchers have proposed defense mechanisms, such as gradient pruning [E], data representation perturbation [F], and differential privacy (DP) [G]. These works are essential. However, they are, in essence, orthogonal to the main theme of this paper, which focuses on contribution evaluation. Nevertheless, these privacy-preserving strategies can be integrated with our method, working collaboratively to enhance data privacy protection. These clarifications will also be added in the revision.
> > >
> > > [C] Zoo: Zeroth order optimization based black-box attacks to deep neural networks without training substitute models. ACM AISec 2017\
> > > [D] Federated learning attacks and defenses: A survey. IEEE Big Data 2022\
> > > [E] Deep leakage from gradients. NeurIPS 2019\
> > > [F] Soteria: Provable defense against privacy leakage in federated learning from representation perspective. CVPR 2021\
> > > [G] Gradient-leakage resilient federated learning. ICDCS 2021

---

### Official Review · Reviewer_a6mr · 2023-07-10

**Soundness:** 3 good
**Presentation:** 3 good
**Contribution:** 3 good
**Rating:** 5
**Confidence:** 2

**Summary:**

The paper introduces a novel approach named Single-round Participants Amalgamation for Contribution Evaluation (SPACE) for efficiently evaluating the contribution of participants in Federated Learning (FL). Accurately evaluating participant contribution has been a challenge in current FL, especially considering cost and scalability. Current methods mostly use the Shapley value, a measure from cooperative game theory, but calculating it is computationally expensive. SPACE, on the other hand, combines two new components - Federated Knowledge Amalgamation and Prototype-based Model Evaluation - to reduce evaluation effort. Federated Knowledge Amalgamation distills information from all local models into the server model in just one round of communication, saving time. Prototype-based Model Evaluation compares server and client prototype similarities, effectively eliminating the dependency on the size of the validation set.

Additionally, SPACE modifies the utility function using a logistic function to better reflect user satisfaction, enhancing the utility function's rationality in real-world settings. Experimental results show that SPACE outperforms current methods in terms of both running time and Pearson's Correlation Coefficient, a measure of correlation strength. The efficacy of SPACE has been tested in various applications, including client reweighting and selection, and consistently demonstrates exceptional performance.

**Strengths:**

- SPACE, appears to be a novel and innovative approach to evaluating the contribution of participants in Federated Learning. The introduction of components like Federated Knowledge Amalgamation and meta-learning using prototypes could potentially provide new avenues for research.
- The method aims to reduce the time and computational resources required for evaluating participant contributions by eliminating dependence on the size of the validation set and enabling evaluation within a single communication round. This efficiency is particularly valuable in Federated Learning, where minimizing communication is crucial due to privacy and resource concerns.

**Weaknesses:**

- Although not specific to this one paper, a recurring trend in federated learning conducts experiments that may overfit to specific use cases. In particular, most papers demonstrate the efficacy of their methods using datasets limited to Federated MNIST and non-iid versions of CIFAR (in this case with only 10 classes). Although these experiments might demonstrate proof of concept, adaptability and effectiveness of the method across a broader range of datasets and scenarios remains unclear. As such, I believe additional experiments with more extensive datasets, or an additional discussion on how this method can be applied to real-world scenarios might help.

**Questions:**

- Are there specific domains or types of applications where SPACE proves particularly effective or ineffective?
- You mention that SPACE enables participant evaluation within a single communication round. What impact does this have on the overall quality and accuracy of the federated learning model?
- How generalizable is the SPACE approach? Could it be adapted to various federated learning architectures and different types of machine learning models?
- Is there a possibility that SPACE could be used in a malicious way, such as inflating a participant's contribution unfairly?
- How does SPACE handle privacy concerns? Does the process of distilling information from local models to a server model pose any risks to data privacy?

**Limitations:**

See weaknesses above

---

> ### Author Rebuttal · Authors · 2023-08-09
>
> We thank the reviewer for the comments and summary of our paper. We have addressed all your questions in the following.
> >  I believe additional experiments with more extensive datasets, or an additional discussion on how this method can be applied to real-world scenarios might help.
>
> Please see "Author Rebuttal by Authors" above.
>
> > Are there specific domains or types of applications where SPACE proves particularly effective or ineffective?
>
> The experimental results indicate that the proposed SPACE approach is effective, especially in a mislabeled data scenario. This is because prototypes obtained from clients with mislabeled data exhibit substantial deviation from the prototypes of the validation set. Consequently, such deviations are efficiently identified by the prototype-based model evaluation approach employed in the proposed method.
>
> > You mention that SPACE enables participant evaluation within a single communication round. What impact does this have on the overall quality and accuracy of the federated learning model?
>
> Our SPACE assesses participants' contributions to the federated learning process rather than directly engaging in model training. Consequently, its implementation does not directly influence the overall quality and accuracy of the federated model. However, when utilized as the weighting mechanism for clients, it contributes positively to the robustness of the federated learning model, as evidenced by the results shown in Figure 2.
>
> > How generalizable is the SPACE approach? Could it be adapted to various federated learning architectures and different types of machine learning models?
>
> Our proposed SPACE is designed to assess participant contributions within the context of horizontal federated learning settings. In such scenarios, participants collectively engage in a federated learning process while sharing a common feature space yet possessing distinct data samples.  Moreover, our SPACE can effortlessly integrate into other deep learning classification models because the two major components, i.e., the information aggregation (Federated Knowledge Amalgamation) and the evaluation protocol (Prototype-based model evaluation), are both model agnostic.
>
> > Is there a possibility that SPACE could be used in a malicious way, such as inflating a participant's contribution unfairly?
>
> If the prototypes of the validation set are exposed, clients could exploit this information to enhance their contributions artificially. A precautionary measure is implemented to counteract the risk of information leakage during the knowledge amalgamation. This involves granting clients access solely to the feature extraction layers of the amalgamated model while excluding the fully-connected layers. This approach facilitates the creation of prototypes without compromising the confidentiality of the distribution underlying the validation set. Another conceivable threat involves clients employing the zeroth-order optimization technique [C] to derive prototypes of disproportionately high contribution. However, this attack can be readily countered by limiting the client's access to the server. Such a limitation is sensible since clients typically do not need to re-calculate their contributions frequently.
>
> > How does SPACE handle privacy concerns? Does the process of distilling information from local models to a server model pose any risks to data privacy?
>
> Indeed, federated learning is susceptible to various attacks [D], mainly when uploading model weights or gradients, as it introduces the potential risk of reconstruction attacks [E]. Such attacks allow adversaries to reconstruct client training data by exploiting predicted confidence values, model parameters, and gradients. To counter these threats, researchers have proposed defense mechanisms, such as gradient pruning [E], data representation perturbation [F], and differential privacy (DP) [G]. These privacy-preserving strategies can be effectively integrated with our method, working collaboratively to enhance data privacy protection. However, it is crucial to emphasize that our current work centers on contribution evaluation, and the topic of attack-defense mechanisms lies beyond the scope of our discussion.
>
> [C] Chen, P. Y., Zhang, H., Sharma, Y., Yi, J., & Hsieh, C. J. (2017, November). Zoo: Zeroth order optimization based black-box attacks to deep neural networks without training substitute models. In Proceedings of the 10th ACM workshop on artificial intelligence and security (pp. 15-26).
>
> [D] Chen, Y., Gui, Y., Lin, H., Gan, W., & Wu, Y. (2022, December). Federated learning attacks and defenses: A survey. In 2022 IEEE International Conference on Big Data (Big Data) (pp. 4256-4265). IEEE.
>
> [E] Zhu, L., Liu, Z., & Han, S. (2019). Deep leakage from gradients. Advances in neural information processing systems, 32.
>
> [F] Sun, J., Li, A., Wang, B., Yang, H., Li, H., & Chen, Y. (2021). Soteria: Provable defense against privacy leakage in federated learning from representation perspective. In Proceedings of the IEEE/CVF conference on computer vision and pattern recognition (pp. 9311-9319).
>
> [G] Wei, W., Liu, L., Wut, Y., Su, G., & Iyengar, A. (2021, July). Gradient-leakage resilient federated learning. In 2021 IEEE 41st International Conference on Distributed Computing Systems (ICDCS) (pp. 797-807). IEEE.

---

> > ### Comment · Reviewer_a6mr · 2023-08-15
> >
> > Thank you for your response. It answers my questions and concerns, so I will be keeping my positive score.

---

### Official Review · Reviewer_vCp6 · 2023-07-27

**Soundness:** 3 good
**Presentation:** 3 good
**Contribution:** 3 good
**Rating:** 6
**Confidence:** 4

**Summary:**

This paper studies the client contribution evaluation problem under federated learning settings. The goal is to achieve more computational and communicational efficient contribution evaluation. The paper proposes Federated Knowledge Amalgamation and Prototype-based Model Evaluation technique for the goal. Federated Knowledge Amalgamation treats client models as teacher models that together train the server model which serves as the student model. This training approach avoids the typical multi-round training of federated learning frameworks, thus reducing the communication costs. Prototype-based Model Evaluation derives the contribution of a client by computing and comparing the prototypes of the client models and the server model, thus reducing both the computation and communication costs of contribution evaluation. Experimental results on MNIST and CIFAR10 datasets confirm the effectiveness of the proposal.

**Strengths:**

1. This paper studies participant contribution evaluation in federated learning which is an important and trending topic.

2. The proposed techniques are intuitive and are effective as shown by experiments on two benchmark datasets.

3. The paper is well written and easy to follow overall.

**Weaknesses:**

While the paper proposes an interesting and empirically effective technique for improving the efficiency of participant contribution evaluation in federated learning, there is no theoretical guarantee on the effectiveness of the proposed technique in terms of measuring the true participant contribution.

Minor presentation issues:
Missing whitespace: "Pandey et al.[34]", "Zhang et al.[64]"


**Questions:**

"For the non-IID scenario, we selected a subset of clients, ranging from 0% to 80%, and assigned them local datasets containing incomplete categories." Which categories are incomplete for each of the clients?

**Limitations:**

As discussed in the paper, the proposed technique relies on "a proper and reliable validation set with accurately labeled data, which can be challenging to achieve in real-world scenarios".

---

> ### Author Rebuttal · Authors · 2023-08-09
>
> We thank the reviewer for the positive review and constructive comments. We provide our responses as follows.
> > While the paper proposes an interesting and empirically effective technique for improving the efficiency of participant contribution evaluation in federated learning, there is no theoretical guarantee on the effectiveness of the proposed technique in terms of measuring the true participant contribution.
>
> Please see "Author Rebuttal by Authors" above.
>
> > Minor presentation issues: Missing whitespace: "Pandey et al.[34]", "Zhang et al.[64]"
>
> Thanks for pointing this out. We would revise it in our manuscript.
>
> > "For the non-IID scenario, we selected a subset of clients, ranging from 0% to 80%, and assigned them local datasets containing incomplete categories." Which categories are incomplete for each of the clients?
>
> Specifically, the non-IID clients only own datasets with partial categories. For instance, a client might be assigned a dataset containing only 3 of the ten classes available in the MNIST dataset. This allocation strategy allows us to create clients with significant heterogeneity in their datasets, resulting in channeling situations in federated learning because the data distribution differs between each client.

---

> > ### Comment · Reviewer_vCp6 · 2023-08-20
> >
> > Thank you for the detailed discussion. It addresses my question. I'm keeping my score since it is already the highest among others.

---

### Author Rebuttal · Authors · 2023-08-09

We would like to thank all the reviewers for their constructive comments. Common questions asked by multiple reviewers would be replied here in a unified manner.

>  Theoretical support of why SPACE leads to an excellent approximation to the original Shapley-based contribution evaluation with re-training

Sorry for the misunderstanding. Actually, our SPACE still applies the Shapley value to calculate each participant's contribution as in previous works. The major contribution of SPACE is that we employ the similarity between prototypes of the server validation set and prototypes formed by clients in a coalition as a utility function for Shapley value calculation instead of the model performance, leading to a lower computation cost. It is worth noting that the theoretical foundation of why we can adopt the similarity function as our utility (for model performance) is that the differences in empirical risk between two distributions are bounded by the divergence between the two distributions when the labeling functions are the same (Theorem 1 in [A]). This finding implies that if the divergence between the client coalition dataset and the validation set is small, then the model trained on the client coalition dataset may obtain high performance on the validation set. In that case, it is reasonable using the divergence to approximate the model performance on the validation set. However, the variation divergence is difficult to compute in practice. Therefore, we intuitively adopt the prototypes' similarity as an alternative to a distribution divergence. The rationale for selecting prototype similarity rests on two crucial considerations. Firstly, it can be easily computed, significantly expediting the evaluation process without necessitating a complete iteration across all samples. Secondly, its privacy-preserving nature renders it suitable for transmission within the framework of federated learning, an aspect that several preceding studies have embraced.

> The experiment is limited in both client numbers and datasets.

In the experiment, we conducted contribution evaluation experiments on the MNIST and CIFAR10 datasets, following the established protocols of previous studies. However, it is important to highlight that the number of clients used in our experiments was limited due to the necessity of calculating the actual Shapley value as the ground truth for evaluation. Specifically, doubling the number of clients from 10 to 20 would introduce an exponential increase in effort by a factor of 1000. Consequently, we performed experiments with a maximum of 10 clients until a more efficient evaluation metric could be proposed. Notably, our proposed SPACE method demonstrated the ability to simultaneously amalgamate information from 100 clients, as demonstrated in the experiment of client selection. During the rebuttal phase, we conduct an additional experiment to include the partial Tiny-ImageNet dataset [B], including 50 categories with ten clients. The results are as follows: The proposed SPACE method achieved satisfactory performance in terms of PCC. Notably, feature alignment is more challenging in such a complex dataset. Thus knowledge amalgamation requires more epochs for convergence, particularly in non-IID scenarios where clients hold data from disjoint classes. The SPACE(Avg) approach, which utilizes FedAvg to obtain the embedding space, may exhibit superior time efficiency for cases with only a few communication rounds. However, the communication benefits from single-round amalgamation still make SPACE the preferred solution when communication cost is a concern. Note that the communication cost shown in the table is calculated as $Comm = 2 \cdot n \cdot R_g \cdot |\theta|$, where $R_g$ denotes the number of communication rounds, $|\theta|$ represents the number of model parameters(MB) and constant of 2 stands for the upload and download of the models. For approaches that require retraining, the number of clients $n$ involves repeated clients during retraining.
* Result on Tiny-ImageNet

|           |  GT       | TMC   | GTG  | DIG-FL  | SPACE(Avg)  | SPACE  |
|---        |---        |---    |---|---|---|---|
| Non-IID   | 0.9082    |  **0.9610**     | 0.7425  | 0.8944   | 0.9175   | 0.9092    |
| Mislabel  | 0.7833    |  0.9233     | 0.7998  | 0.8405   | **0.9293**   | 0.9256    |
| Times(s)  | 84994     |  69212      | 98855   | 266      | **252**      | 453   |
| Comm(MB)  | 2639200   |  2097600    | 8000    | 8000     | 8000     | **400**   |

[A] Ben-David, S., Blitzer, J., Crammer, K., Kulesza, A., Pereira, F., & Vaughan, J. W. (2010). A theory of learning from different domains. Machine learning, 79, 151-175.

[B] Le, Y., & Yang, X. (2015). Tiny imagenet visual recognition challenge. CS 231N, 7(7), 3.

---

### Decision · Program_Chairs · 2023-09-21

**Decision:**

Accept (poster)

**Comment:**

The majority of reviewers lean towards accepting the paper. vCp6 and a6mr raise the issue of generality, which they deem adequately addressed in the rebuttal. a6mr also questions on privacy concerns, but accepts the understanding that the key focus of this paper is on contribution evaluation, which is complementary. AsTD raises the issue of malicious users, which is sufficiently addressed in the rebuttal and authors are encouraged to include the discussion in the revised version. AA9d questions the assumptions in SPACE and is satisfied by the responses. HEwC requests more information on communication efficiency, where the results in the response are important and may be included in the revised version. Overall, the meta-reviewer agrees with the reviewers that the paper may be accepted. Authors are encouraged to include the above revisions.